



# Enhancement and validation of a state-of-the-art global hydrological model H08 (v.bio1) to simulate second-generation herbaceous bioenergy crop yield

Zhipin Ai[1], Naota Hanasaki[1], Vera Heck[2], Tomoko Hasegawa[3], Shinichiro Fujimori[4]

[1]Center for Climate Change Adaptation, National Institute for Environmental Studies, 16-2, Onogawa, Tsukuba 305-8506, Japan

[2]Potsdam Institute for Climate Impact Research, Telegraphenberg A 31, Potsdam 14473, Germany

[3]Department of Civil and Environmental Engineering, Ritsumeikan University, 56-1, Toji-in Kitamachi, Kita-ku, Kyoto 603-8577, Japan

[4]Department of Environmental Engineering, Kyoto University, Building C1-3, C-cluster, Kyoto-Daigaku-Katsura, Nishikyo-ku, Kyoto 615-8504, Japan

*Correspondence to*: Zhipin Ai (ai.zhipin@nies.go.jp)

**Abstract.** The bioenergy crop yield is a critical determinant of the bioenergy potential for various stringent climate change mitigation scenarios. Currently, the bioenergy crop yield is usually determined from a limited number of simulations. However, reliable yield simulation remains a challenge at the global scale. Here, through parameter calibration and algorithm improvement, we enhanced a state-of-the-art global hydrological model (H08) to simulate the bioenergy yield
from dedicated the herbaceous bioenergy crops *Miscanthus* and switchgrass. Site-specific evaluations showed that the enhanced H08 had the ability to simulate yield for both *Miscanthus* and switchgrass, with the calibrated yields being well within the ranges of the observed yield. Independent country-specific evaluations further confirmed the performance of the enhanced H08. Using this improved model, we found that unconstrained irrigation more than doubled the yield of the rainfed condition, but reduced the water use efficiency (WUE) by 29% globally. With irrigation, the yield in dry climate zones can
exceed the rainfed yields in tropical climate zones. Nevertheless, due to the low water consumption in tropical areas, the highest WUE was found in tropical climate zones, regardless of whether the crop was irrigated.



## 1 Introduction

The bioenergy with carbon capture and storage (BECCS) technology enables the production of energy without carbon emissions, while sequestering carbon dioxide from the atmosphere, producing negative emissions. Therefore, BECCS is

considered an important technology in the push to achieve the 2-degree climate target (Smith et al., 2015). With ambitious climate policies, the demand for bioenergy could reach 200–300 EJ per year, based on recent predictions (Rose et al., 2013). Second-generation bioenergy crops, such as *Miscanthus* and switchgrass, are generally regarded as a dedicated bioenergy source due to the high yield potential and their lack of direct competition with food production (Beringer et al., 2011). This is because *Miscanthus* and switchgrass are rhizomatous perennial $C_4$ grasses, which have a high photosynthesis efficiency

(Trybula et al., 2015). Satisfactory simulation of the yield of *Miscanthus* and switchgrass is therefore critically important to reliably project bioenergy production levels and the potential of BECCS (Trybula et al., 2015; Heck et al., 2016).

Although the history of simulating *Miscanthus* or switchgrass yield is relatively short, much progress at site or national scale has been made in the last two decades. For example, Clifton-Brown et al. (2000) developed a radiation-driven model that

could be coupled with a geographic information system (GIS) to estimate the yield of *Miscanthus* in Ireland. The model parameters, including the leaf area index and light extinction coefficient, were determined based on field measurements in 1994 and 1995 in south central Ireland. The reported yield varied from 16 to 26 Mg ha$^{-1}$ yr$^{-1}$. It was noted that the model neglected water stress. Later, this model was scaled up to the continental level to estimate the *Miscanthus* yield within the European Union (Clifton-Brown et al., 2004). The upscaling was based on the ordinary kriging interpolation of the original

study for Ireland. The new model was called MISCANMOD. The results showed that the maximum yield in Europe varied from 13 Mg ha$^{-1}$ yr$^{-1}$ in Finland to 25.8 Mg ha$^{-1}$ yr$^{-1}$ in Belgium under rainfed conditions. With improved consideration of the effects of water stress and air temperature on radiation-use efficiency, Hastings et al. (2009) improved MISCANMOD and developed a Fortran-based model called MISCANFOR. Recently, the focus has shifted to the modification of the existing crop module to adapt to the physiological characteristics of *Miscanthus* or switchgrass. For example, Trybula et al. (2015)

improved the Soil & Water Assessment Tool (SWAT) model by modifying the leaf area development curve and revising the nutrient uptake response to environmental stress for *Miscanthus* and upland switchgrass based on field experiment data at the agronomy center of Purdue University from 2009 to 2011. Another example is the modification of the Agricultural Production Systems Simulator (APSIM) model. Ojeda et al. (2017) improved the APSIM model by introducing a new parameterization scheme for *Miscanthus* and switchgrass by calibrating with broad site-specific data gathered in the US.


Simulations of both *Miscanthus* and switchgrass yield at the global scale are still scarce. Beringer et al. (2011) extended the Lund-Potsdam-Jena managed Land (LPJmL) model (Bondeau et al., 2007) for lignocellulosic energy crops to three crop functional types: tropical trees, temperate trees, and $C_4$ grasses. The $C_4$ grasses were parameterized to represent *Miscanthus* and switchgrass. The simulated $C_4$ grass yield had a satisfactory performance through simple evaluations with *Miscanthus*

(five observations and 20-32 predictions) and switchgrass (three observations and six predictions). Heck et al. (2016) reevaluated the performance of the LPJmL model in bioenergy crop yield simulation with more observations (32 for *Miscanthus* and 28 for switchgrass). Kang et al. (2014) established a framework and conducted a global study for simulating switchgrass yield. This framework was based on the High-Performance Computing Environmental Policy Integrated Climate (HPC-EPIC) model developed by Nichols et al. (2011), which enables the spatiotemporally extensive application of a well-

established EPIC model (Williams et al., 1984). Using the framework, Kang et al. (2014) estimated the global distribution of switchgrass yield. Finally, Li et al. (2018b) enhanced the global vegetation model ORCHIDEE to simulate lignocellulosic bioenergy crops.





Although much progress has been made, several limitations remain. First, the systematic calibration and extensive validation of models need to be improved to be used at a global scale. For example, the modelling work on LPJmL was calibrated manually (Beringer et al., 2011; Heck et al., 2016), and the simulation work based on HPC-EPIC was calibrated systematically but lacked independent validation and further model inter-comparisons (Kang et al., 2014). Second, to the best of our knowledge, the effect of irrigation on the yield of both *Miscanthus* and switchgrass, particularly their water use efficiency (WUE) in different climate zones at the global scale, has not been well studied. Third, simulations have generally been performed for either *Miscanthus* (Clifton-Brown et al., 2000, 2004; Hastings et al., 2009) or switchgrass (Kang et al., 2014), with only few studies of both (Trybula et al., 2015; Ojeda et al., 2017; Beringer et al., 2011; Li et al., 2018b). Fourth, previous studies have generally been conducted at the regional or continental scale (Clifton-Brown et al., 2000, 2004; Hastings et al., 2009; Trybula et al., 2015; Ojeda et al., 2017); few have been conducted at the global scale (Beringer et al., 2011; Heck et al., 2016; Kang et al., 2014; Li et al., 2018b). Fifth, despite their importance, the key parameters for *Miscanthus* and switchgrass and their differences have only been well documented in a few studies (Trybula et al., 2015). Sixth, except for the LPJmL model, few models can simulate the bioenergy crop yield with full consideration of human activities in the water sector, such as irrigation, reservoir operation, and water withdrawal. To fill these gaps and limitations, we collected up-to-date site-specific and country-specific yield data for both *Miscanthus* and switchgrass to conduct model calibration and validation. We then simulated the yield of both *Miscanthus* and switchgrass with a state-of-the-art global hydrological model, H08 (see details in section 2.1).

The following sections of this paper will: 1) describe the default biophysical process of the crop module in H08, 2) explain the proposed enhancement of H08 for *Miscanthus* and switchgrass, 3) evaluate the enhanced performance of the model in simulating yields for *Miscanthus* and switchgrass, and 4) illustrate the effect of irrigation on the yield, water consumption, and WUE of *Miscanthus* and switchgrass.

## 2 Materials and methods

### 2.1 H08 and its crop module

H08 is a global hydrological model that has the primary aim of addressing the impacts of major human activities, such as irrigation and reservoir operation, on the global hydrological cycle. It was initially designed with six sub-modules, including land surface water and energy exchange, river routing, reservoir operation, crop growth, environmental flow, and anthropogenic water withdrawal (Hanasaki et al., 2008a, 2008b). H08 can simulate the basic natural and anthropogenic hydrological process as well as crop growth at a spatial resolution of 0.5° and at a daily interval. The six sub-modules are coupled in a unique way. The land surface module can simulate the main water cycle components, such as evapotranspiration and runoff. The former is used in the crop module, and the latter is used in the river routing and environmental flow modules. The agricultural water demand simulated by the crop module and the streamflow simulated by the river routing and reservoir operation module finally enter into the withdrawal module. A graphical diagram illustrating these coupled relationships can be found in Hanasaki et al. (2008b).

Figure 1 shows the basic biophysical process of the crop module in H08. The biomass accumulation is based on Monteith et al. (1977). The crop phenology development is based on daily heat unit accumulation theory. The harvest index is used to partition the grain yield. Regulating factors, including water and air temperature, are used to constrain the yield variation (see supplementary material for information on the algorithms). The crop module can simulate the potential yield, crop calendar, and irrigation water consumption for 18 crops, including barely, cassava, cotton, peanut, maize, millet, oil palm, potato, pulses, rape, rice, rye, sorghum, soybean, sugar beet, sugarcane, sunflower, and wheat. The parameters for these





crops were taken from those of the SWAT model. To better reflect the agronomy practice, H08 divides each simulation cell into four sub-cells: rainfed, single-irrigated, double-irrigated, and other (i.e., non-agricultural land uses).

### 2.2 Enhancement of H08 for *Miscanthus* and switchgrass

To establish its ability to address perennial bioenergy crops, the crop sub-module of H08 was enhanced to include functions for the second-generation bioenergy crops *Miscanthus giganteus* and the switchgrass *Panicum virgatum*. The first use of H08 to simulate the bioenergy crop yield was reported in an impact assessment of the effects of BECCS on water, land, and ecosystem services (Yamagata et al., 2018). This first bioenergy crop implementation was realized using three steps. First, the crop parameters for *Miscanthus* and switchgrass were adopted based on the settings from the SWAT model 2012 version

(Arnold et al., 2013). Second, because both *Miscanthus* and switchgrass are perennial, the potential heat unit was set as unlimited. Maturity was defined by either suffering from an autumn freeze (i.e., the air temperature is below the minimum temperature for growth) or exceeding the maximum of 300 continuous days of growth. Third, and most importantly, the model performance for the simulated bioenergy crop yield was not validated at all.

To make the model and bioenergy yield simulations more reliable, we extensively modified the bioenergy crop sub-model. First, we changed the leaf area development curve by adopting the potential heat unit (Hun) and leaf area related parameters (dpl1 and dpl2) proposed by Trybula et al. (2015). The potential heat unit can determine both the total cropping days and the leaf development. Here, we set it at 1,830 and 1,400 degrees for *Miscanthus* and switchgrass, respectively, as recommended by Trybula et al. (2015) based on their field observations. The dpl1 and dpl2 parameters (see Table 1), which were used for

determining the leaf development curve, were also changed to the values suggested by Trybula et al. (2015). This modification substantially changed the original heat unit index (Ihun) and the development of the leaf area index curve. Second, we modified the algorithm for water stress that was used to regulate the radiation use efficiency. We took the ratio of actual evapotranspiration to potential evapotranspiration as the water stress factor for any point in the simulation, similar to the description of the soil moisture deficit used in other studies (Anderson et al., 2007; Yao et al., 2010). Third, we

conducted parameter calibrations based on a series of simulations. The calibration process is presented in section 2.5, and the finalized parameter settings are given in Table 1 and section 3.1. Fourth, we added as an output item the water consumption of *Miscanthus* and switchgrass to analyze the water consumption and WUE in the crop sub-module. Fifth, we fixed the bug in the original code. For definitions and the functions of the above parameters, such as Hun, dpl1, dpl2, and Ihun, please see the algorithm descriptions in the supplementary material.


### 2.3 Model input data

The WATCH-Forcing-Data-ERA-Interim (WFDEI) global meteorological data (Weedon et al., 2014) from 1979 to 2016 were used in all simulations. The WFDEI data were based on the methodology used for WATer and global CHange (WATCH) forcing data by utilizing ERA-Interim global reanalysis data. The data cover the whole globe at a spatial

resolution of 0.5°. Eight daily meteorological variables (air temperature, wind speed, air pressure, specific humidity, rainfall, snowfall, and downward shortwave and longwave radiation) were used to run H08.

### 2.4 Yield data

To independently calibrate and validate the performance of H08 in simulating the bioenergy yield, we collected and

compiled up-to-date site-specific and country-specific yield data from both observations and simulations (Clifton-Brown et al., 2004; Searle and Malins, 2014; Heck et al., 2016; Kang et al., 2014; Li et al., 2018a). For *Miscanthus*, the yield data used covered 64 sites (observed) and eight counties (simulated). For switchgrass, the yield data used covered 55 sites (observed) and 16 countries (simulated). A map showing the locations of the sites is presented in Fig. 2. The data sites were



predominantly distributed in Europe and the US. Detailed lists of the sites from which the yields of *Miscanthus* and
switchgrass were reported are documented in Tables s1 and s2 in the supplementary material, respectively.

### 2.5 Simulation and analysis

To reduce the inter-annual variation and avoid extreme cold air temperatures in early spring, we conducted simulations with
the mean meteorological conditions within the period 1979–2016 (38 years). Two simulations were run with different
purposes. The first simulation was conducted with the land surface module and the crop module without irrigation to
calibrate and validate both the original and enhanced H08 models. The second simulation was also conducted with the land
surface module and the crop module but with irrigation to investigate the yield potential under irrigated conditions with the
enhanced H08. It should be noted that irrigation in this study means uniform unconstrained irrigation.

We conducted a calibration with the most sensitive parameters, such as radiation use efficiency (be), maximum leaf area
index (blai), base temperature (Tb), maximum daily accumulation of temperature (Hunmax), and minimum temperature for
planting (TSAW). The specific parameter ranges and steps set in the calibration process are shown in Table 2. In total, 1,944
simulations were conducted for *Miscanthus* and switchgrass to test all combinations of the parameter sets. More information
on how these parameters affect the model can be found in the algorithm description section in the supplementary material.
The best parameter sets were selected using two steps: first, the lowest root mean square error (RMSE), and second, the
highest correlation coefficient (R) of the simulated and observed yields within the lowest RMSE domain. To conduct the
calibration and validation, the observed site-specific data were used to calibrate the model, and the simulated country-
specific data were used to validate the model. The site-specific data covered different latitudes, with ranges from 7.0°S to
56.8°N for *Miscanthus* and 28.45°N to 51.8°N for switchgrass. The collected country-specific data cover the three different
models: MISCANMOD, HPC-EPIC, and LPJmL. This analysis provided an opportunity to illustrate yield-latitude
relationships as well as the limitations and performance of the model. In addition, we introduced the Köppen climate
classification into the source code to provide possible climate-specific analyses.

### 3 Results and discussion

### 3.1 Parameter calibration

Based on the optimal RMSE (4.68 and 3.16 Mg ha$^{-1}$ yr$^{-1}$ for *Miscanthus* and switchgrass, respectively) and R (0.67 and 0.53
for *Miscanthus* and switchgrass, respectively), we finalized the parameter set as shown in Table 1. The simulations presented
in the table are for rainfed conditions, because the few sites that were irrigated were excluded from the analysis. The
radiation-use efficiency was set at 38 and 22 (g MJ$^{-1}$ × 10) for *Miscanthus* and switchgrass, respectively. These values are
similar to previous reports. For example, a value of 41 (g MJ$^{-1}$ × 10) for Miscanthus and 17 (g MJ$^{-1}$ × 10) for switchgrass
was recommended by Trybula et al. (2015). The base temperature was calibrated to be 8 and 10°C for *Miscanthus* and
switchgrass, respectively. The base temperature is sensitive to the crop growing days. Ranges from 7 to 10°C for *Miscanthus*
and from 8 to 12°C for upland switchgrass were suggested by Trybula et al. (2015). The calibrated values are within the
above ranges. The maximum leaf area indices were calibrated at 11 and 8 for *Miscanthus* and switchgrass, respectively;
these values were identical to those suggested by Trybula et al. (2015).

### 3.2 Site-specific performance of enhanced H08

An overview of the performance of the enhanced H08 is provided in Fig. 3. It can be seen that the performance of the
enhanced H08 was improved over that of the original H08, with the overestimation and underestimation tendencies having
been successfully fixed. Points in a scatter plot comparing the simulated yield from the enhanced H08 with the observed
yield were well distributed along the 1:1 line. More detailed site-specific results are shown in Fig. 4a (*Miscanthus*) and Fig.





4b (switchgrass). To depict the uncertainties in the observed yield, the minimum and maximum observed yields were added as error bars in Fig. 4. It was found that the simulated yields were within or close to the range of the observed yield, and the relative errors for both *Miscanthus* and switchgrass were well distributed along the 0 line.


### 3.3 Country-specific performance of enhanced H08

Figure 5 compares the yield simulated by the enhanced H08 with the collected independent country-specific yields simulated by MISCANMOD (Clifton-Brown et al., 2004), HPC-EPIC (Kang et al., 2014), and LPJmL (Heck et al., 2016). Here, the yield was simulated under rainfed conditions. For *Miscanthus*, the correlation coefficient of the yield simulated by H08 and

MISCANMOD in the scatter plot (Fig. 5e) was 0.65. A t-test showed that the correlation was not significant at the 0.01 level ($p<0.01$) (it was significant at the 0.08 level, $p<0.08$). For consistency with the yield collected by MISCANMOD, any countries with a yield less than 10 Mg ha$^{-1}$ yr$^{-1}$ were excluded from the analyses. Also, the land available for calculation was set as 10% of the pastureland and cropland. For switchgrass, the correlation coefficient of the yield simulated by H08 and HPC-EPIC in the scatter plot (Fig. 5f) was 0.80. A t-test showed that the correlation was significant at the 0.01 level. This

indicates that the spatial pattern of the yield simulated by H08 was similar to that of HPC-EPIC. For example, high yields were found in Brazil, Colombia, Mozambique, and Madagascar, while low yields were found in Australia and Mongolia by both models.

*Miscanthus* and switchgrass are not distinguished in LPJmL, and we therefore compared the mixed (mean, *Miscanthus* and

switchgrass) yield of *Miscanthus* and switchgrass simulated by H08 and the C$_4$ grass yield simulated by LPJmL. The correlation coefficient of the yield simulated by H08 and LPJmL in the scatter plot (Fig. 5g) was 0.84, much higher than that obtained with MISCANMOD and HPE-EPIC. A t-test showed that the correlation was significant at the 0.01 level. The difference was mainly due to Colombia, Sudan, Mozambique, and Mexico, which are located in tropical zones. The difference in these countries was generally equal to the range of H08. For example, the yield in Colombia simulated by

LPJmL was equal to the *Miscanthus* yield simulated by H08 (upper error bar). It was difficult to determine which model performed better due to the lack of observed data in tropical zones. This also indirectly indicated the relatively large uncertainty of the existing simulations in tropical zones (Kang et al., 2014).

The differences in model structure, use of specific algorithms, and period covered by the input climate data can induce

differences in the yield simulated by MISCANMOD, HPC-EPIC, LPJmL, and H08. With regard to model structure, MISCANMOD uses a Kriging interpolation method to derive the spatial yield from the original site yield, whereas H08, LPJmL, and HPC-EPIC use grid-based calculations. With regard to the specific algorithms used, the water stress used to regulate radiation-use efficiency is quite different among the models. The periods of climate data used as an input are 1960–1990, 1980–2010, 1982–2005, and 1979–2016 for the MISCANMOD, HPC-EPIC, LPJmL, and H08 models, respectively.

Thus, the climate data used for MISCANMOD are quite different from those used by the other models.

### 3.4 Spatial distribution of the simulated yield under rainfed and irrigated conditions

Figure 6 shows the global yield distribution of *Miscanthus* and switchgrass. Under rainfed conditions, high yields are distributed in eastern US, Brazil, southern China, Africa, and Southeast Asia. To evaluate the response of yield to irrigation,

we compared two simulations under rainfed and irrigated conditions. As shown in Figs. 6c and 6d, unconstrained irrigation greatly increased yields, especially for areas in arid regions such as the western US, southern Europe, northeastern China, India, southern Africa, the Middle East, and coastal Australia. At the global scale, the increases (excluding the area with a polar climate) were 22.2 (from 16.8 to 39.0) Mg ha$^{-1}$ yr$^{-1}$ and 8.8 (from 7.4 to 16.2) Mg ha$^{-1}$ yr$^{-1}$ for *Miscanthus* and switchgrass, respectively, indicating that irrigation more than doubles the yield under rainfed conditions. The spatial





distribution of yield increased due to the irrigation simulated by H08 being very similar to that simulated by LPJmL (Beringer et al., 2011). At the continental scale (e.g., Europe), the yield increase was mainly located in southern Europe, consistent with the findings obtained using MISCANMOD (Clifton-Brown et al., 2004). The yield response to irrigation for switchgrass was weaker than that for *Miscanthus* (see Figs. 6b and 6d). This might be due to a smaller dependency on water for switchgrass compared with *Miscanthus* (Mclsaac et al., 2010). *Miscanthus* growth has been reported to have a high water

requirement due to the high yield, large leaf area index, and long growing season (Mclsaac et al., 2010; Lewandowski et al. 2003). As a result, the *Miscanthus* yield is strongly influenced by water availability, and an annual rainfall of 762 mm $yr^{-1}$ is thought to be suitable for growth (Heaton et al., 2019). However, the precipitation in most locations is below this level, especially arid and semi-arid regions (see Fig. s1 in the supplementary material). Therefore, irrigation plays a critical role in ensuring the optimum bioenergy crop yield in arid and semi-arid regions, especially for *Miscanthus*.


**3.5 Effects of irrigation on yield, water consumption, and WUE in different climate zones**

    Climate is one of the main physical constraints of crop growth and yield. Figure 7a shows the mean yield for *Miscanthus* and switchgrass in four different Köppen climate zones (see Fig. s2 in the supplementary material). For *Miscanthus*, a tropical climate (including the northern part of South America, central Africa, Southeast Asia, and southern India) produced the

highest average yield of 31.8 Mg $ha^{-1}$ $yr^{-1}$. A temperate climate (including the eastern US, Europe, southern China, and the southern part of South America) produced the second highest average yield of 20.6 Mg $ha^{-1}$ $yr^{-1}$. Dry and continental climate zones had similar average yields of 7.6 and 7.3 Mg $ha^{-1}$ $yr^{-1}$, respectively. For switchgrass, a tropical climate had the highest yield, averaging 11.2 Mg $ha^{-1}$ $yr^{-1}$. For the other three climate types, the average yields averaged 9.4, 4.4, and 4.4 Mg $ha^{-1}$ $yr^{-1}$ for the temperate, continental, and dry climate zones, respectively. As shown in Fig. 7a, irrigation greatly increased the

yield, especially in dry climate zones, which had the largest yield increases of 46.6 and 17.5 Mg $ha^{-1}$ $yr^{-1}$ for *Miscanthus* and switchgrass, respectively. In contrast, irrigation had a relatively weak effect on yield in the tropical climate zone.

    Figure 7b shows the water consumption for both *Miscanthus* and switchgrass. The annual mean water consumption for *Miscanthus* was around 602 mm $yr^{-1}$ for the tropical climate zone (with a high yield of 31.8 Mg $ha^{-1}$ $yr^{-1}$), whereas it was 157

mm $yr^{-1}$ for a dry climate (with a low yield of 7.6 Mg $ha^{-1}$ $yr^{-1}$) under rainfed conditions. Under irrigated conditions, the largest increases in water consumption were 1,676 and 1,124 mm $yr^{-1}$ for *Miscanthus* and switchgrass in dry climate zones, respectively. With such a large amount of irrigation, the yield in a dry climate zone can exceed that in a tropical climate zone under rainfed conditions. This highlights the yield-water tradeoff effects.

Figure 7c shows the WUE, which is defined in this study as the ratio of yield to water consumption. The WUE of *Miscanthus* in a tropical climate was 52.8 kg DM $ha^{-1}$ $mm^{-1}$ $H_2O$, and 48.4, 45.5, and 36.1 kg DM $ha^{-1}$ $mm^{-1}$ $H_2O$ in dry, temperate, and continental climate zones under rainfed conditions. The WUE values of switchgrass were 35.2, 29.8, 26.6, and 26.3 kg DM $ha^{-1}$ $mm^{-1}$ $H_2O$ in temperate, tropical, dry, and continental climate zones under rainfed conditions, respectively. The WUE values for *Miscanthus* was higher than those for switchgrass, which is inconsistent with values in

previous reports (VanLoocke et al., 2012). With irrigation, the WUE decreased for both *Miscanthus* and switchgrass in all climate zones. Globally, excluding the area with a polar climate, the decreases were 13.1 (from 47.6 to 34.5) kg DM $ha^{-1}$ $mm^{-1}$ $H_2O$ and 8.7 (from 28.8 to 20.1) kg DM $ha^{-1}$ $mm^{-1}$ $H_2O$ for *Miscanthus* and switchgrass, respectively, indicating a reduction in the mean WUE values for *Miscanthus* and switchgrass of up to 29%. This is consistent with the current global WUE trend for crops, which is high for rainfed croplands but low for irrigated croplands. However, the general magnitude of

this relationship changes if the site or regional scale is considered based on reports for wheat in Syria (Oweis et al., 2000) or for wheat and maize in the North China Plain (Mo et al., 2005).



### 3.6 Limitations

There were several limitations in this study that need to be addressed. First, the bioenergy crop yield simulated by H08 did
not include constraints due to nutrients, pesticides, or agronomy management. Also, the effects of $CO_2$ fertilizer and
technological advancements were not considered in the current simulations. Second, our simulation was conducted with
historical meteorological drivers. Therefore, variations in yield in future climate scenarios under different representative
concentration pathways need to be examined. Third, the current irrigation levels were input to represent uniform
unconstrained irrigation. Further evaluations need to consider the availability of renewable water sources, and planetary
boundaries of land, food, and water (Heck et al., 2018).

### 4 Conclusion

In this study, we enhanced the ability of the H08 global hydrological model to simulate the yield of a dedicated second-
generation bioenergy crop. The enhanced H08 generally performed well in globally simulating the yield of both *Miscanthus*
and switchgrass. To the best of our knowledge, this study is the first attempt to successfully enable a global hydrological
model, with consideration of human activity, to simulate the yield of a bioenergy crop. The enhanced model could be a good
tool for the future assessment of the bioenergy-water relations. We also quantified the effects of irrigation on yield, water
consumption, and WUE for both *Miscanthus* and switchgrass in different climate zones. Irrigation more than doubled the
yield in all areas under rainfed conditions and reduced the WUE by 29%. With irrigation, the yield in a dry climate could
exceed that in a tropical climate under rainfed conditions. However, due to the low water consumption in tropical areas, the
highest WUE was generally in tropical climate zones, regardless of whether the crop was irrigated.

Currently, observed yield data for second-generation bioenergy crops, such as *Miscanthus* and switchgrass, are scarce at the
global scale due to the lack of large-scale plantations (Beringer et al., 2011). Decisions regarding bioenergy production are
largely based on a limited number of simulations (Trybula et al., 2015). Moreover, future projections of bioenergy potential
are usually strongly dependent on the historical baseline yield estimated using biophysical models (Heck et al., 2016). Our
study therefore contributed to both the bioenergy crop yield and simulation research communities. Compared with earlier
studies, our study made several important improvements. First, rather than using an approximation for C₄ grass to represent
*Miscanthus* and switchgrass in the LPJmL model, our enhanced H08 model simultaneously simulated the yields for
*Miscanthus* and switchgrass at the global scale. Second, we conducted both a site-specific calibration and independent
country-specific evaluation with more observed data (as can be seen in Table 3) and predicted data (from the MISCANMOD,
HPC-EPIC, and LPJmL models). Third, because of the importance of transparent parameter selection as underscored by
Trybula et al. (2015), we disclosed the parameters set for both *Miscanthus* and switchgrass. Fourth, we further investigated
the differences in yield, water consumption, and WUE for both *Miscanthus* and switchgrass among different climate zones.
Finally, except for existing models, such as the LPJmL model, our enhanced H08 model provides new ways to evaluate the
future impacts of human activities, such as irrigation, reservoir operation, and water withdrawal, on the large-scale
establishment of bioenergy plantations.

*Code and data availability.* The code of the model used in this study is archived on Zenodo
(https://zenodo.org/record/3521407#.XbjZqiXTZMB) under Creative Commons Attribution 4.0 International License.
Technical information about the H08 model and the input dataset are available from the following website: http:
//h08.nies.go.jp.

*Competing interests.* The authors declare that they have no conflict of interest.




*Author contribution.* Naota Hanasaki designed this study. Zhipin Ai collected the data, developed the model code, and performed the simulations. Zhipin Ai prepared the manuscript, with contributions and comments from Naota Hanasaki, Vera Heck, Tomoko Hasegawa, and Shinichiro Fujimori.

*Acknowledgments.* This study was funded by the Environment Research and Technology Development Fund (S-14) of the Environmental Restoration and Conservation Agency, Japan.





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





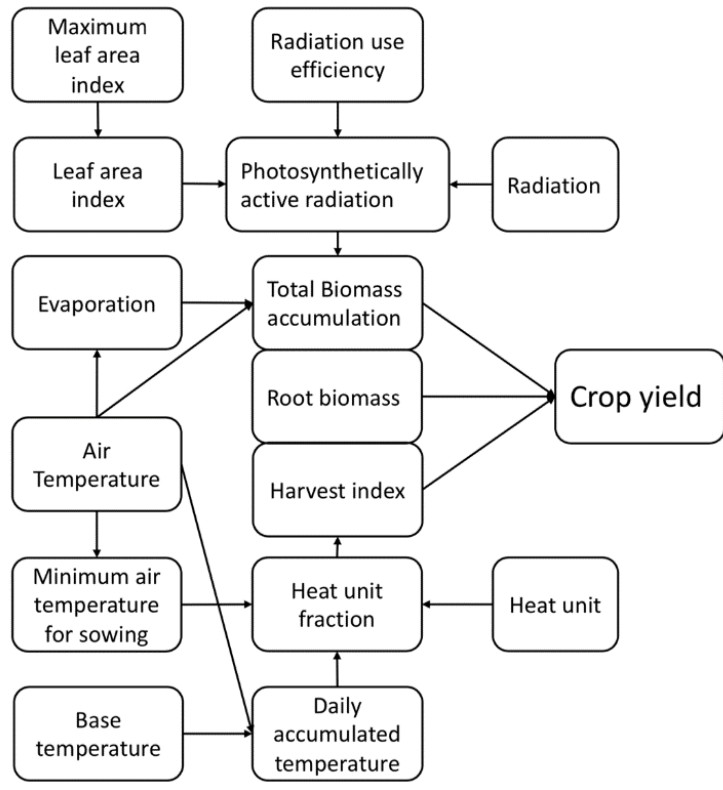

Fig. 1 Schematic diagram showing the basic biophysical process of the crop module in the H08 model.



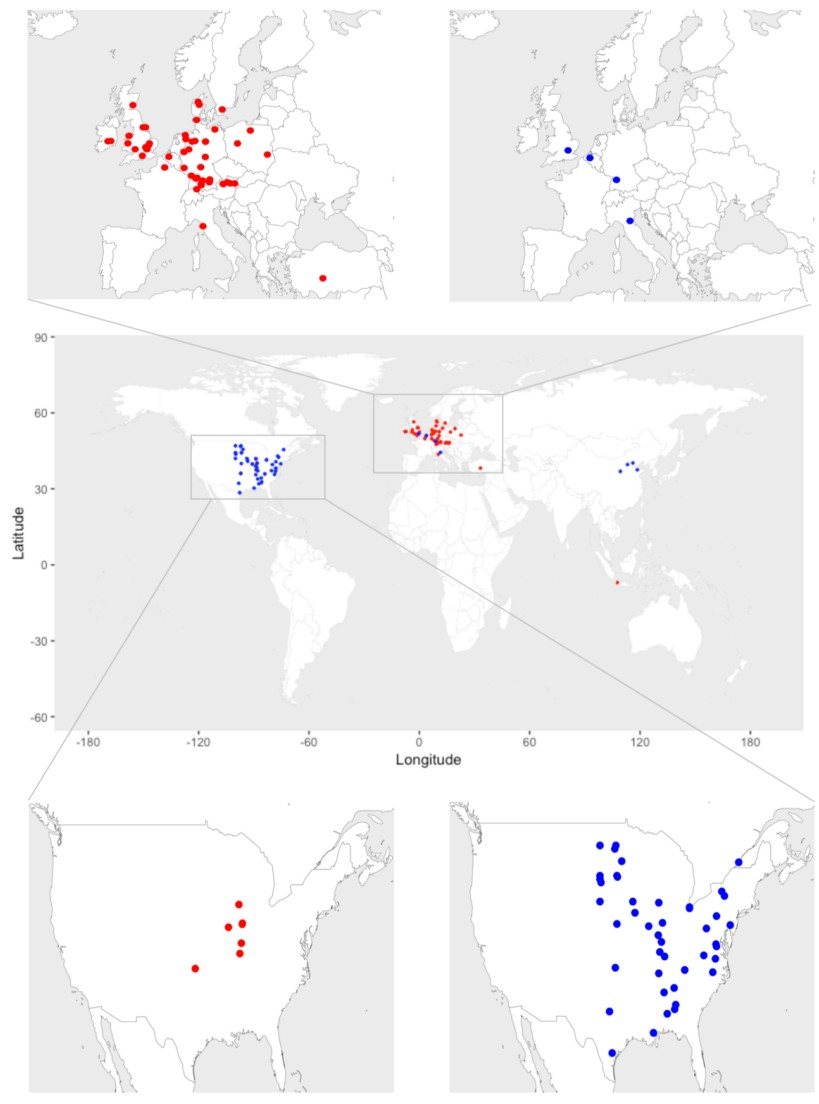

**Fig. 2 Map showing the locations of the *Miscanthus* (red dots) and switchgrass (blue dots) sites.**






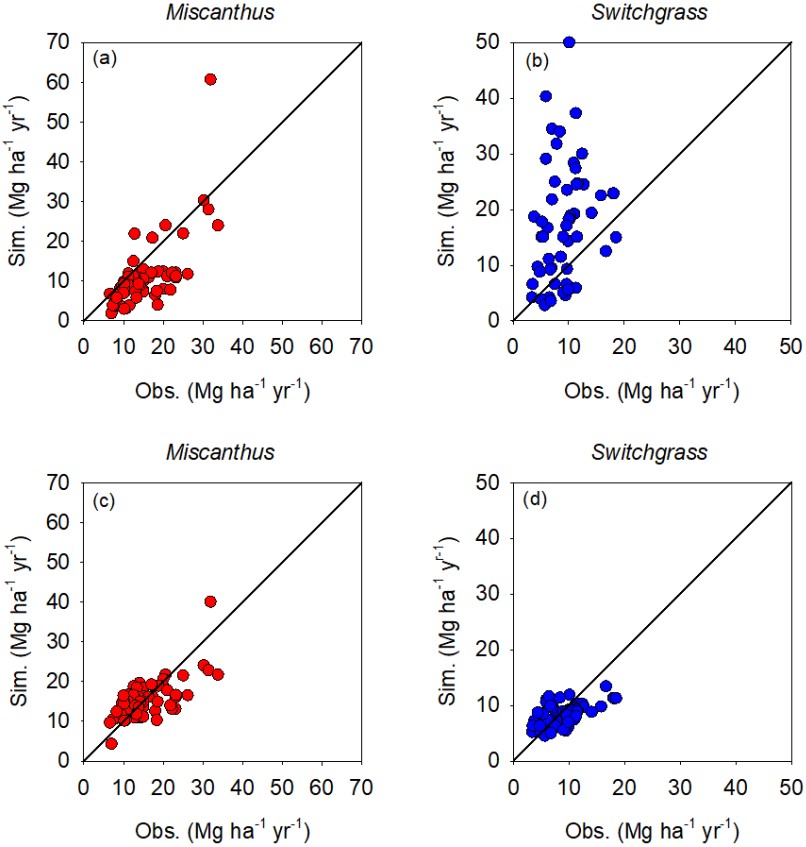

**Fig. 3** Overall comparison of the simulated (Sim.) and observed (Obs.) yields for *Miscanthus* and switchgrass, respectively. The simulated yields in (a) and (b) are from the original H08 model, whereas those in (c) and (d) are from the enhanced H08 model. The black line is the 1:1 line.



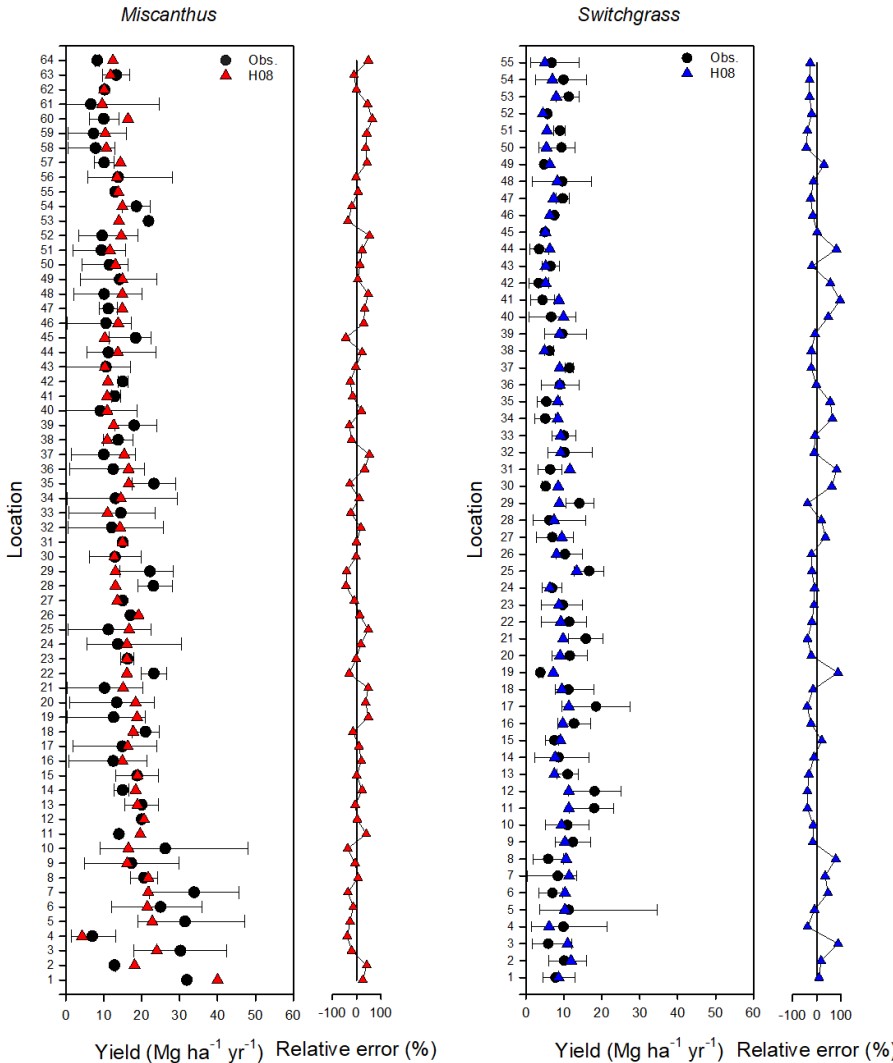


**Fig. 4 Site-specific performance (presented with latitude increasing from the bottom of the vertical axis) and relative error of the simulated yield obtained using the enhanced H08 model compared with the observed yields for *Miscanthus* and switchgrass. The longitude and latitude of each location for *Miscanthus* and switchgrass are given in Tables s1 and s2, respectively. Obs. means the observed mean yield. The error bar represents the range of the**

**observed minimum and maximum yield, respectively.**




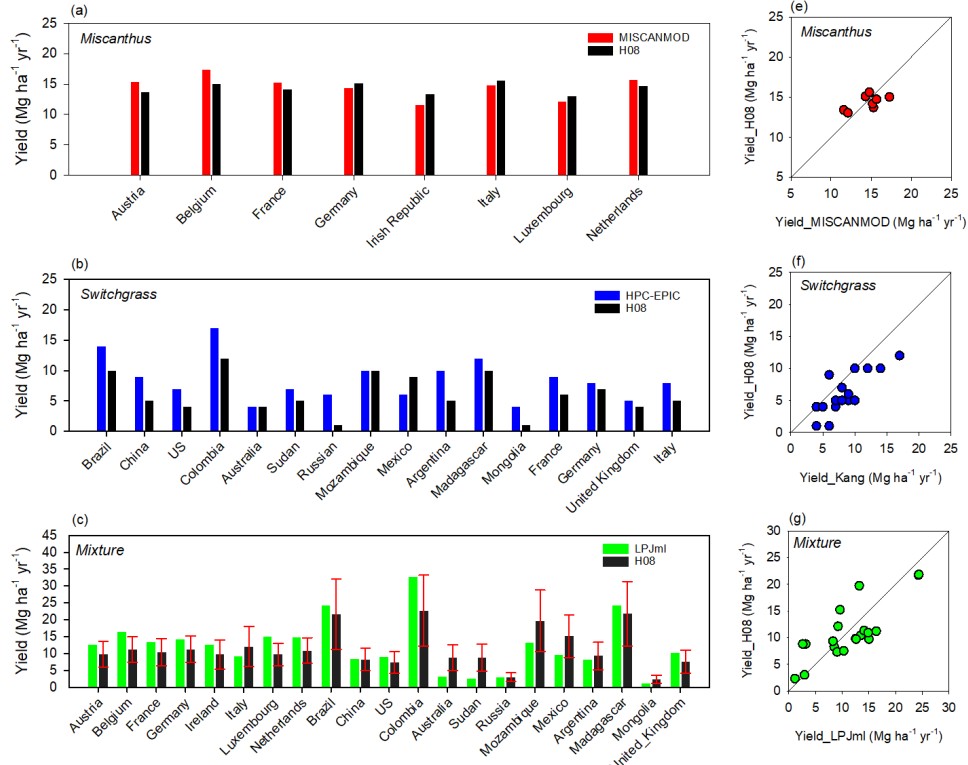

**Fig. 5 An independent country-specific comparison of the simulated yield by the enhanced H08 model with those of three other models (MISCANMOD, HPC-EPIC, and LPJmL) for *Miscanthus* (a, e), switchgrass (b, f), and their combination (c, g), respectively. The H08 in (c) indicates the average yield of *Miscanthus* and switchgrass, and the upper and lower error bars in (c) represent the yields for *Miscanthus* and switchgrass, respectively.**


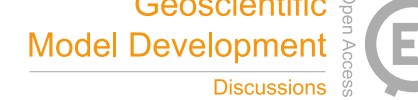

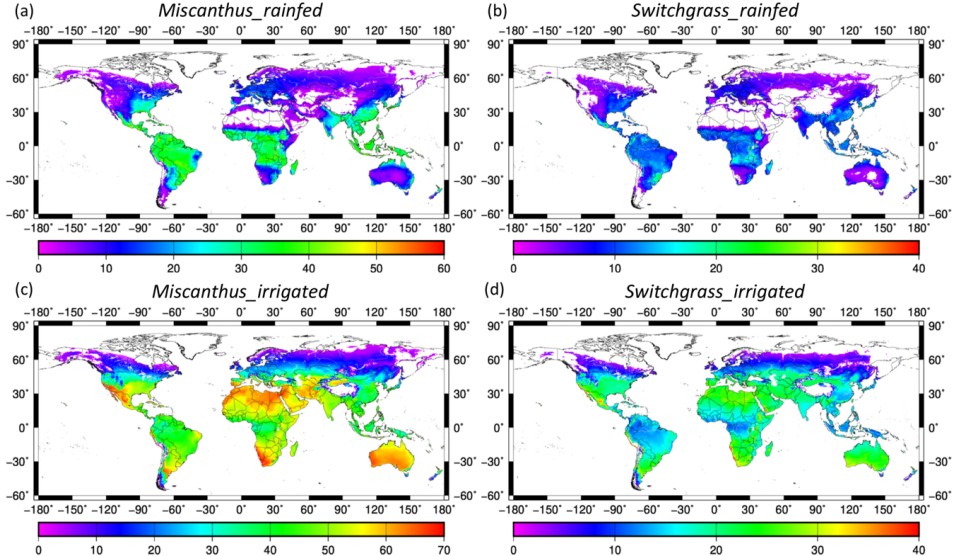

**Fig. 6 Spatial distributions of the simulated yields (exceeds 2 Mg ha$^{-1}$ yr$^{-1}$) for** *Miscanthus* **(a, c) and switchgrass (b, d) under rainfed (a, b) and irrigated (c, d) conditions, respectively. The unit for the legend is Mg ha$^{-1}$ yr$^{-1}$.**






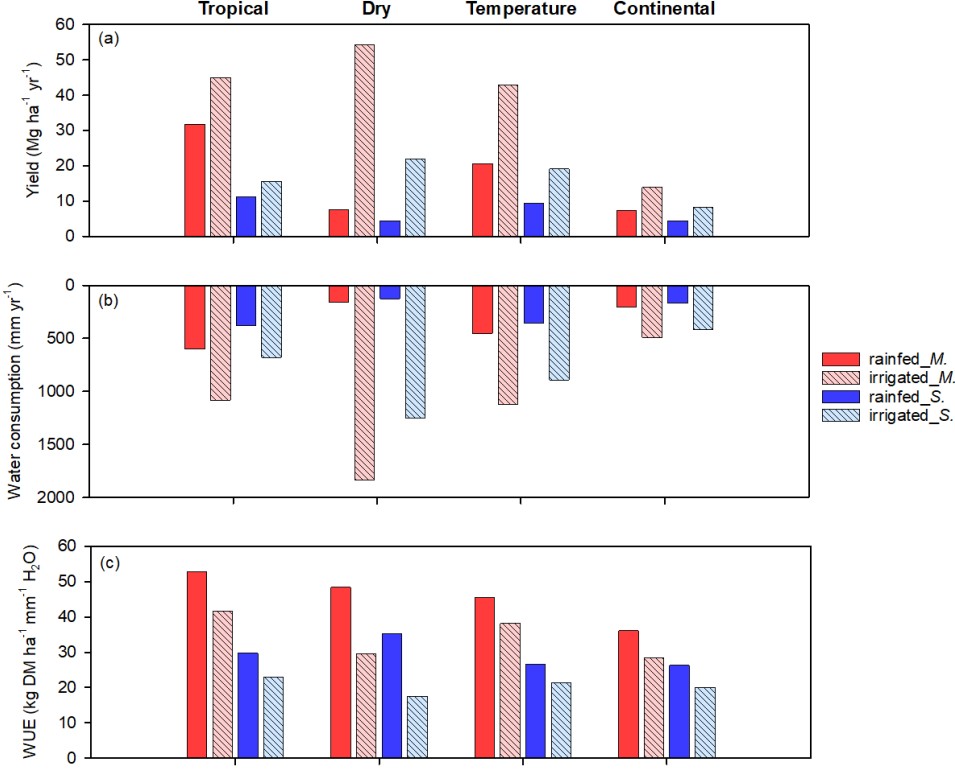

**Fig. 7 Variations in the average yield (a), crop water consumption (b), and water use efficiency (WUE) (c) for *Miscanthus* and switchgrass under rainfed and irrigated conditions in four different Köppen climate zones (tropical, dry, temperate, and continental climates) based on meteorology data collected from 1979 to 2016. The abbreviations M. and S. in the legend denote *Miscanthus* and switchgrass, respectively.**






**Table 1. Parameters set in the enhanced H08 model.**


| Bioenergy crop | Parameter | Value | Source |
|---|---|---|---|
| *Miscanthus* | Hun | 1,830 | Trybula et al., (2015) |
| | be | 38 | Calibrated |
| | To | 25 | Trybula et al., (2015); Hastings et al., (2009) |
| | Tb | 8 | Calibrated |
| | blai | 11 | Calibrated |
| | dlai | 1.1 | Trybula et al., (2015) |
| | dpl1 | 10.1 | Trybula et al., (2015) |
| | dpl2 | 45.85 | Trybula et al., (2015) |
| | rdmx | 3 | Trybula et al., (2015) |
| | Hunmax | 11.5 | Calibrated |
| | TSAW | 8.0 | Calibrated |
| Switchgrass | Hun | 1,400 | Trybula et al., (2015) |
| | be | 22 | Calibrated |
| | To | 25 | Trybula et al., (2015) |
| | Tb | 10 | Calibrated |
| | blai | 8 | Calibrated |
| | dlai | 1 | Trybula et al., (2015) |
| | dpl1 | 10.1 | Trybula et al., (2015) |
| | dpl2 | 40.85 | Trybula et al., (2015) |
| | rdmx | 3 | Trybula et al., (2015) |
| | Hunmax | 15.5 | Calibrated |
| | TSAW | 8.0 | Calibrated |



**Table 2. Parameter ranges and steps for calibration simulations.**

| Bioenergy crop | Parameter | Range | Step | Unit | Reference |
|---|---|---|---|---|---|
| *Miscanthus* | be | (30, 40) | 2 | g MJ$^{-1}$ × 10 | Clifton-Brown et al., (2000); van der Werf et al., (1992); Beale and Long, (1995); Heaton et al., (2008); Trybula et al., (2015) |
| | blai | (9, 11) | 1 | m$^2$ m$^{-2}$ | Heaton et al., (2008); Trybula et al., (2015) |
| | Tb | (7, 9) | 1 | ℃ | Beale et al., (1996); Trybula et al., (2015) |
| | Hunmax | (11.5, 16.5) | 1 | ℃ | H08 Endogenous variable |
| | TSAW | (8, 10) | 1 | ℃ | H08 Endogenous variable |
| Switchgrass | be | (12, 22) | 2 | g MJ$^{-1}$ × 10 | Heaton et al., (2008); Madakadze et al., (1998); Trybula et al., (2015) |
| | blai | (6, 8) | 1 | m$^2$ m$^{-2}$ | Trybula et al., (2015); Giannoulis et al., (2016); Madakadze et al., (1998); Heaton et al., (2008) |
| | Tb | (8, 10) | 1 | ℃ | Trybula et al., (2015) |
| | Hunmax | (11.5, 16.5) | 1 | ℃ | H08 Endogenous variable |
| | TSAW | (8, 10) | 1 | ℃ | H08 Endogenous variable |

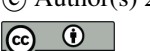



**Table 3. Comparison of the number of sites or countries used for calibration and validation with previous global studies. Single asterisks denote numbers of sites, and the double asterisk denotes a number of countries.**


| Study | Scale | *Miscanthus* | | | | Switchgrass | | | |
|---|---|---|---|---|---|---|---|---|---|
| | | Calibration | | Validation | | Calibration | | Validation | |
| | | Observed | Simulated | Observed | Simulated | Observed | Simulated | Observed | Simulated |
| Beringer et al. (2011) | Global | - | - | 5[*] | 20[*] to 32[*] | - | - | 3[*] | 6[*] |
| Kang et al. (2014) | Global | - | - | - | - | 52[*] | - | - | - |
| Heck et al. (2016) | Global | - | - | 32* | - | - | - | 28* | - |
| This study | Global | 64[*] | - | | 32[**] | 55[*] | - | | 40[**] |