# Peer review of "Enhancement and validation of a state-of-the-art global hydrological model H08 (v.bio1) to simulate second-generation herbaceous bioenergy crop yield"

_Geoscientific Model Development, 2019_

## Referee Comment (RC1) · Anonymous Referee #1 · 20 Dec 2019

The manuscript evaluates yield potential and water use efficiency of bioenergy crops Miscanthus and switchgrass at global scale using H08 model. The study reads as an adaptation of Trybula etal 2017 to H08 model and calibrating model for field cites from multiple countries to expand to global scale simulation. My major concerns are whether the model parameterization from Trybula et al for Midwest US weather suitable for global scale (specific comments below) and is the general water stress accounting model is reasonable for perennial bioenergy crops. The manuscript is well organized and easy to read.

Specific Comments: 1. Selection of hydrologic model H08: Authors mentions H08 as a state-of the art model multiple times in the manuscript. Of the available hydrology models, what makes H08 state of the art model? Additional discussion would be helpful 2. The study primarily focuses on bioenergy production potential and water use efficiency. The water use efficiency is estimated using simple scenario analysis of with and without water stress (through irrigation). I am wondering whether specific crop model could be better suited for such analysis rather than hydrologic model 3. It will be nice if authors list the goals and objectives of the manuscript 4. Enhancement of H08 for miscanthus and switchgrass: a. Authors chose potential heat units to maturity as 1830 and 1400 for miscanthus and switchgrass respectively based on Tryubla et al (2015). The HU for Trybula et al was estimated for continental climate with winter crop senescence. Is this valid for other climates? b. The water stress representation is similar to many hydrological models with stress as direct function of actual ET/potential ET. The crop water stress tolerance and impact on biomass production is crop specific. Some additional discussion of the stress functions for specific crops will be interesting since WUE is major focus of the study. 5. 2.5 Simulation and analysis: why authors chose to reduce interannual variability in temperature? 6. Results and discussion: I appreciate authors efforts to list all model parameters and compare parameters and simulation results with literature. The optimal RMSE and R performance after calibration is on lower side especially for switchgrass. 7. In section 3.2 authors claim "over estimation and underestimation tendencies having been successfully fixed" for H08 model, this seems to be a strong claim considering low performance indicators. I agree the improved version is better than original H08 simulations. Figure 4 simulated yield relative error well distributed along the 0 line, the range is -100 to +100 and the x axis relatively small and that makes the lines look closer to 0 relative error. 8. Section 3.3: " the land available for calculation was set as 10% of the pastureland and cropland" any specific justification for choosing this?

---

## Referee Comment (RC2) · Anonymous Referee #2 · 25 Dec 2019

Comments on "Enhancement and validation of a state-of-the-art global hydrological model H08 (v.bio1) to simulate second-generation herbaceous bioenergy crop yield" by Ai et al.

The ms shows modeling implementations and results of global simulations of switch-grass and Miscanthus yields, and effects of irrigation in the simulations. I appreciate the courageous work to validate global simulations of energy crops, however, the ms doesn't provide any original scientific insight; it just adopts modeling information from SWAT model. Also, the ms doens't incorporate various insight from researches on energy crops, for example, suitable energy crop species in tropics, importance of fertilizer applications on marginal land, etc.

I have several concerns about the ms, and I do not find the ms ready for publication in present form. I must recommend publication in another outlet specific to bioenergy researches (e.g. Biomass and Bioenergy, GCB Bioenergy, . . .) after substantial revision.

Major concerns:

1. This ms considers only two lignocellulosic herbaceous energy crops, switchgrass and Miscanthus. These species are mainly considered as energy crops for temperate and continental climate zone. For the global simulations of second-generation herbaceous energy crops, authors need to address additional species in the modeling like Napier grass, sugarcane/energycane which are suitable for growing in tropics (Surendra et al. 2018). Also, consideration of woody species like Eucalyptus in tropics and SRC in boreal is recommended in the analysis of yield comparison with LPJmL and/or ORCHIDEE-MICT-BIOENERGY models as they already simulate.

2. Many studies show sensitivities on the yields, and requirement, even limited amount, of N and/or P fertilizer for these crops, particularly on switchgrass (e.g. Wullschleger et al. 2010, Hong et al. 2014, Ashworth et al. 2016). SWAT model already contains implementations of the process of fertilizers applications. This is critical point.

3. The simulated yields by the HO8 should be compared with simulation using SWAT model with the same climate forcing to validate the adopted implementations from the model described in the ms.

4. The simulation uses climatology (1997-2016 average) for the input, however, to validate the model results with field observations, I would recommend using annual forcing data in the simulations for the corresponding years. In addition, only WFDEI forcing is used in the simulations. There is no supporting information on the rational of using the forcing date over other data set, like S14FD (Iizumi et al. 2017), CRU JRA.

(Harris et al. 2019). At minimum, a revised manuscript must address the limitations.

5. It is critically needed to compared/validate the simulated yields using irrigation with field observations/experiments if data would be available.

Minor comments:

L16: Bauer et al. 2018 could be a "recent predictions".

L61: Miguez et al. 2012 also considers both species.

L62-64: Is it really few studies available? Nair et al. 2012 reviews and shows many models already available for the simulation.

L154: I found no description on simulation algorithm of parameter ensemble in the supporting materials.

Fig.3: Is it possible to disaggregate the figure with each climate zones?

Table s1, s2: I would recommend adding mean and SD of the observed yields in the tables.

L190: Too small sample size (N = 8).

L192: I don't understand the reason of excluding that yield below 10 Mg ha-1 yr-1.

L199: Does the result of Fig 5g change if the higher yield from Miscanthus or switch-grass calculated by H08 are used in the comparison?

L203: Is the discrepancy related to the modeling assumption of double cropping in the region? In practice, double or triple cropping can be done in the tropics for the herbaceous crops.

L215: I recommend comparing simulated results between SWAT and H08 using same forcing data.

References Ashworth et al., 2016, Agriculture, Ecosystems and Environment,

http://dx.doi.org/10.1016/j.agee.2016.09.041

Bauer et al., 2018, Climatic Change, https://doi.org/10.1007/s10584-018-2226-y

Harris et al., 2019, http://dx.doi.org/10.5285/13f3635174794bb98cf8ac4b0ee8f4ed

Hong et al., 2014, BioEnergy Research, https://doi.org/10.1007/s12155-014-9484-yu

Iizumi et al., 2017, Journal of Geophysical Research, https://doi.org/10.1002/2017JD026613

Miguez et al., 2012, Global Change Biology, https://doi.org/10.1111/j.1757-1707.2011.01150.x

Nair et al., 2012, Global Change Biology, https://doi.org/10.1111/j.1757-1707.2012.01166.x

Surendra et al., 2018, Bioresource Technology, https://doi.org/10.1016/j.biortech.2017.12.044

Wullschleger et al., 2010, Agron. J., https://doi.org/10.2134/agronj2010.0087

---

## Short Comment (SC1) · 31 Dec 2019

I am working on using satellite remote sensing to estimate the crop yield. I am interested in this paper since it aims to improve the hydrologic model to estimate the bioenergy crop yields on the global scale. Although I am new to crop models, I feel this paper is easy to understand and provide some insights for my own work. I would appreciate authors share more details about the model parameters and compare simulation results with past models. Then, the audiences will have more direct feelings about the improvement of the model.

---

## Author Comment (AC1) · 29 Jan 2020

**We would like to thank Reviewer 1 for the insightful comments which helped us a lot to improve the manuscript. Below you will find our response to each comment. The reviewer's comments are shown in italic and the responses are shown in bold letters.**

*# Reviewer1*

*[R1-M1] The manuscript evaluates yield potential and water use efficiency of bioenergy crops Miscanthus and switchgrass at global scale using H08 model. The study reads as an adaptation of Trybula et al 2017 to H08 model and calibrating model for field cites from multiple countries to expand to global scale simulation. My major concerns are whether the model parameterization from Trybula et al for Midwest US weather suitable for global scale (specific comments below) and is the general water stress accounting model is reasonable for perennial bioenergy crops. The manuscript is well organized and easy to read.*

**Response: We thank you for taking time to review this manuscript. Our primary goal is to improve H08 to better simulate the second-generation herbaceous bioenergy crop yield, which is a critical variable in estimating global bioenergy potential. The study of Trybula et al. (2015) provides us valuable information for our work.**

**Regarding your first concern, as you can find in Table 1, our finalized parameters contains both reference from Trybula et al. (2015) and our own calibration. In the revised manuscript, we identified the site's climate zone and added such information in Fig. 3 and Fig. 4. As can be seen in Fig. 4, the performance of our simulation is good, which demonstrates this parameterization scheme has the capability of global application. Regarding your second concern, as shown in Fig. 3 and Fig. 4, the simulation results agree well with observation hence the water stress method is valid for perennial bioenergy crops.**

*Specific Comments:*

*[R1-S1] Selection of hydrologic model H08: Authors mentions H08 as a state-of the art model multiple times in the manuscript. Of the available hydrology models, what makes H08 state of the art model? Additional discussion would be helpful*

**Response: The strength and uniqueness of the H08 is its considerations of human activities (or water management) including reservoir operation, aqueduct water transfer, seawater desalination, and water abstraction for irrigation, industry, and municipality (Hanasaki et al., 2008a, 2008b, 2010, 2018a, 2018b). Indeed, H08 contains most extensive water management options among the global hydrological models [WaterGAP (Alcamo et al., 2003; Döll et al., 2003, 2012, 2014), LPJmL (Gerten et al., 2004; Rost et al., 2008; Biemans et al., 2011), PCR-GLOBWB (van Beek et al., 2011; Wada et al., 2011, 2014), WBM-plus (Vörösmarty et al., 1989; Wisser et al., 2010), HiGW-MAT (Pokhrel et al., 2012a, b, 2015), and others]**

**We have added text from line 81 to line 83.**

*[R1-S2] The study primarily focuses on bioenergy production potential and water use efficiency. The water use efficiency is estimated using simple scenario analysis of with and without water stress (through irrigation). I am wondering whether specific crop model could be better suited for such analysis rather than hydrologic model*

**Response: The direct goal of this study is to add a function to H08 to simulate the second-generation herbaceous bioenergy crop yield. Our long-term goal is to investigate the bioenergy-water tradeoffs at global scale making full use of the capability of H08 as a global hydrological model. We agree with you that it might be better to use specific crop model, if one wish to investigate the water use efficiency at site or watershed scale, but this is not the case of this study.**

*[R1-S3] It will be nice if authors list the goals and objectives of the manuscript*

**Response: Thanks for your comment. We added them at the beginning of the last paragraph in the introduction section.**

*[R1-S4] Enhancement of H08 for miscanthus and switchgrass: a. Authors chose potential heat units to maturity as 1830 and 1400 for miscanthus and switchgrass respectively based on Tryubla et al (2015). The HU for Trybula et al was estimated for continental climate with winter crop senescence. Is this valid for other climates? b. The water stress representation is similar to many hydrological models with stress as direct function of actual ET/potential ET.*

*The crop water stress tolerance and impact on biomass production is crop specific. Some additional discussion of the stress functions for specific crops will be interesting since WUE is major focus of the study.*

**Response: Thanks for your insightful comments. Regarding comment a, adopting the HU identified by Trybula et al. (2015) to the entire globe seems valid based on the results from Fig. 4. HU determines the cropping duration, hence critically sensitive to the results. The good agreement between simulation and observation at various climatic zones indicate the validity of HU setting. Regarding comment b, we agree with you that water stress tolerance should be crop specific (such as implication from Hastings et al., 2009). Ideally speaking, the relationship between water stress and yield must be carefully monitored and modeled. However, such monitoring and modeling studies are not available, as far as we know, and if available, it must be quite challenging to extend the model for global application. This is why many earlier studies (such as Anderson et al., 2007; Yao et al., 2010) and other generic/global models like MISCANMOD (Clifton-Brown et al., 2004), SWAT (Neitsch et al., 2011), and LPJml (Bondeau et al., 2007) all adopt a crop-unspecific formulation. For example, in MISCANMOD, crops are affected by water stress when soil moisture deficit is more than 0.3, Similarly, we adopted a value of 0.25 in this study.**

**Finally, as mentioned earlier, although shown in text, estimation of WUE is not the final objective of this model development. Rather this paper is intended to describe a model extension of H08 to properly simulate biomass crop yield. The results of WUE are shown since it is calculatable with a stand-alone model.**

**We added the abovementioned discussion in lines 273-275, and lines 288-289.**

*[R1-S5] 2.5 Simulation and analysis: why authors chose to reduce interannual variability in temperature?*

**Response: Thanks for your comment. As noted by the manual of H08 (http://h08.nies.go.jp/h08/manual.html), we used the mean meteorological data**

to avoid the extreme cold air temperatures in early spring. Following the comment of Reviewer 1 and Reviewer 2, in the revised manuscript, we conducted the simulations annually and also revised corresponding text.

*[R1-S6] Results and discussion: I appreciate authors efforts to list all model parameters and compare parameters and simulation results with literature. The optimal RMSE and R performance after calibration is on lower side especially for switchgrass.*

Response: We presented the parameters set and references in Table 1 and Table 2. Yes, the yield is on lower side especially for switchgrass (*Panicum virgatum*) after calibration. We think this is more accurate compared with the lower observed yield shown in Fig. 4b.

*[R1-S7] In section 3.2 authors claim "over estimation and underestimation tendencies having been successfully fixed" for H08 model, this seems to be a strong claim considering low performance indicators. I agree the improved version is better than original H08 simulations. Figure 4 simulated yield relative error well distributed along the 0 line, the range is -100 to +100 and the x axis relatively small and that makes the lines look closer to 0 relative error.*

Response: First, we appreciate your agreement on the improvement of the simulation. Second, based on your comment, we have enlarged the x axis in Fig. 4 and modified the corresponding text.

*[R1-S7] Section 3.3: "the land available for calculation was set as 10% of the pastureland and cropland" any specific justification for choosing this?*

Response: Thanks for your comment. The reason is to make the comparison meaningful since this land set was reported in MISCANMOD country yield estimation (Clifton-Brown et al., 2004).

Reference:

Alcamo, J., Döll, P., Henrichs, T., Kaspar, F., Lehner, B., Rösch, T., and Siebert, S.: Development and testing. of. the WaterGAP 2 global model of water use and availability, Hydrolog. Sci. J., 48, 317–337, https://doi.org/10.1623/hysj.48.3.317.45290, 2003.

Biemans, H., Haddeland, I., Kabat, P., Ludwig, F., Hutjes, R. W. A., Heinke, J., von. Bloh, W., and Gerten, D.: Impact of reservoirs on river discharge and irrigation water supply dur- ing the 20th century, Water Resour. Res., 47, W03509, https://doi.org/10.1029/2009wr008929, 2011.

Bondeau, A., Smith, P. C., Zaehle, S., Schaphoff, S., Lucht, W., Cramer, W., Gerten, D., Lotze-Campen, H., Müller, C., Reichstein, M., and Smith, B.: Modelling the role of agriculture for the 20th century global terrestrial carbon balance, Glob. Change Biol., 13, 679-706, https://doi:10.1111/j.1365-2486.2006.01305.x, 2007.

Clifton-Brown, J. C., Stampfl, P. F., and Jones, M. B.: Miscanthus biomass production for energy in. Europe and its potential contribution to decreasing fossil fuel carbon emissions, Glob. Change Biol., 10, 509-518, https://doi:10.1111/j.1529- 8817.2003.00749.x, 2004.

Döll, P., Kaspar, F., and Lehner, B.: A global hydrological model for deriving water. availability indicators: model tuning and validation, J. Hydrol., 270, 105–134, https://doi.org/10.1016/S0022- 1694(02)00283-4, 2003.

Döll, P., Hoffmann-Dobrev, H., Portmann, F. T., Siebert, S., Eicker, A., Rodell, M., Strassberg, G., and Scanlon, B. R.: Impact of water withdrawals from groundwater and surface water on continental water storage variations, J. Geodyn., 59–60, 143–156, https://doi.org/10.1016/j.jog.2011.05.001, 2012.

Döll, P., Müller Schmied, H., Schuh, C., Portmann, F. T., and Eicker, A.: Global-scale. assessment of groundwater depletion and related groundwater abstractions: combining hydro- logical modeling with information from well observations and GRACE satellites, Water Resour. Res., 50, 5698–5720, https://doi.org/10.1002/2014wr015595, 2014.

Gerten, D., Schaphoff, S., Haberlandt, U., Lucht, W., and Sitch, S.: Terrestrial. vegetation and water balance - hydrological evaluation of a dynamic global vegetation model, J. Hydrol., 286, 249–270, https://doi.org/10.1016/j.jhydrol.2003.09.029, 2004.

Hanasaki, N., Kanae, S., Oki, T., Masuda, K., Motoya, K., Shirakawa, N., et al. (2008a). An integrated model for the assessment of global water resources—Part 2: Applications and assessments. Hydrology and Earth System Sciences, 12(4), 1027–1037. https://doi.org/10.5194/hess-12-1027-2008

Hanasaki, N., Kanae, S., Oki, T., Masuda, K., Motoya, K., Shirakawa, N., et al. (2008b). An integrated model for the assessment of global water resources—Part 1: Model description and input meteorological forcing. Hydrology and. Earth System Sciences, 12(4), 1007–1025. https:// doi.org/10.5194/hess-12-1007-2008

Hanasaki, N., Yoshikawa, S., Pokhrel, Y., & Kanae, S. (2018). A global hydrological. simulation to specify the sources of water used by humans. Hydrology and Earth System Sciences, 22(1), 789–817. https://doi.org/10.5194/hess-22-789-2018.

Hanasaki, N., Inuzuka, T., Kanae, S., & Oki, T. (2010). An estimation of global virtual. water flow and sources of water withdrawal for major crops and livestock products using a global hydrological model. Journal of Hydrology, 384(3–4), 232–244. https://doi.org/10.1016/j. jhydrol.2009.09.028

Hanasaki, N., Yoshikawa, S., Pokhrel, Y., & Kanae, S. (2018). A quantitative. investigation of the thresholds for two conventional water scarcity indicators using a state-of-the-art global hydrological model with human activities. Water Resources Research, 54, 8279–8294. https://doi.org/10.1029/ 2018WR022931

Heaton, E. A., Dohleman, F. G., and Long, S. P.: Meeting US biofuel goals with less land: the potential of Miscanthus, Glob. Change Biol., 14, 2000-2014, https://doi:10.1111/j.1365-2486.2008.01662.x, 2008.

Miguez, F. E., Maughan, M., Bollero, G. A., & Long, S. P. (2012). Modeling spatial and dynamic variation in growth, yield, and yield stability of the bioenergy crops Miscanthus× giganteus and Panicum virgatum across the conterminous United States. Gcb Bioenergy, 4(5), 509-520. https://doi.org/10.1111/j.1757-1707.2011.01150.x

Pokhrel, Y., Hanasaki, N., Koirala, S., Cho, J., Yeh, P. J. F., Kim, H., Kanae, S., and. Oki, T.: Incorporating Anthropogenic Water Regulation Modules into a Land Surface Model, J. Hydrometeorol., 13, 255–269, https://doi.org/10.1175/jhm-d-11-013.1, 2012a.

Pokhrel, Y. N., Hanasaki, N., Yeh, P. J. F., Yamada, T. J., Kanae, S., and Oki, T.: Model. estimates of sea-level change due to anthropogenic impacts on terrestrial water storage, Nat. Geosci., 5, 389–392, https://doi.org/10.1038/ngeo1476, 2012b.

Pokhrel, Y. N., Koirala, S., Yeh, P. J. F., Hanasaki, N., Longuevergne, L., Kanae, S., and Oki, T.: Incorporation of groundwater pumping in a global Land Surface Model with the representation of human impacts, Water Resour. Res., 51, 78–96, https://doi.org/10.1002/2014wr015602, 2015.

van Beek, L. P. H., Wada, Y., and Bierkens, M. F. P.: Global monthly water stress: 1. Water balance and water availability, Water Resour. Res., 47, W07517, https://doi.org/10.1029/2010wr009791, 2011.

VanLoocke, A., Twine, T. E., Zeri, M., & Bernacchi, C. J.: A regional comparison of water use efficiency for miscanthus, switchgrass and maize,

Agric. Forest Meteorol., 164, 82-95, http://dx.doi.org/10.1016/j.agrformet.2012.05.016, 2012.

Vörösmarty, C. J., Moore III, B., Grace, A. L., Gildea, M. P., Melillo, J. M., Peterson, B. J., Rastetter, E. B., and Steudler, P. A.: Continental scale models of water balance and fluvial transport: an application to South America, Global Biogeochem. Cy., 3–3, 241–265, https://doi.org/10.1029/GB003i003p00241, 1989.

Wada, Y., van Beek, L. P. H., Viviroli, D., Dürr, H. H., Weingartner, R., and Bierkens, M. F. P.: Global monthly water stress: 2. Water demand and severity of water stress, Water Resour. Res., 47, W07518, https://doi.org/10.1029/2010wr009792, 2011.

Wada, Y., Wisser, D., and Bierkens, M. F. P.: Global modeling of withdrawal, allocation and consumptive use of surface water and groundwater resources, Earth Syst. Dynam., 5, 15–40, https://doi.org/10.5194/esd-5-15-2014, 2014.

Wisser, D., Fekete, B. M., Vörösmarty, C. J., and Schumann, A. H.: Reconstructing. 20th century global hydrography: a contribution to the Global Terrestrial Network-Hydrology (GTN-H), Hydrol. Earth Syst. Sci., 14, 1–24, https://doi.org/10.5194/hess- 14-1-2010, 2010.

Thank you very much for your comments and suggestions.

Sincerely yours,

Zhipin Ai (on behalf of the co-authors)

---

## Author Comment (AC2) · 29 Jan 2020

We would like to thank Reviewer 2 for the insightful comments which helped us a lot to improve the manuscript. Below you will find our response to each comment. The reviewer's comments are shown in italic and the responses are shown in bold letters.

*#Reviewer 2*

*Comments on "Enhancement and validation of a state-of-the-art global hydrological model H08 (v.bio1) to simulate second-generation herbaceous bioenergy crop yield" by Ai et al. The ms shows modeling implementations and results of global simulations of switchgrass and Miscanthus yields, and effects of irrigation in the simulations. I appreciate the courageous work to validate global simulations of energy crops, however, the ms doesn't provide any original scientific insight; it just adopts modeling information from SWAT model. Also, the ms doens't incorporate various insight from researches on energy crops, for example, suitable energy crop species in tropics, importance of fertilizer applications on marginal land, etc. I have several concerns about the ms, and I do not find the ms ready for publication in present form. I must recommend publication in another outlet specific to bioenergy researches (e.g. Biomass and Bioenergy, GCB Bioenergy,...) after substantial revision.*

**Response: We appreciate the detailed and insightful comments. We well take all your concerns. We totally agree with you that our study does not include every aspect of bioenergy and our crop model is simple. Still we believe this study contains substantial original scientific insights. Also this study tries to shed light on irrigation which is one of the factor often missing in large-scale studies. We also think this paper is within the scope in GMD since this study proposes a functional extension of the well-established global hydrological model called H08.**

**We found the reviewer's main concern is the scientific insight. Therefore, we would like to give specific explanations below.**

**The primary targeted energy crop here is the so-called second-generation herbaceous bioenergy crop, namely Miscanthus and switchgrass. The primary**

spatial scale here is global. Under this circumstance, we give quite comprehensive review of previous reports on yield simulations of Miscanthus and switchgrass and emphasized the progress of global-scale studies. To do so, we identified the limitations and gaps in previous studies as follows:

1) The systematic calibration and extensive validation of models need to be improved at a global scale. For example, the modelling work on LPJmL was calibrated manually (Beringer et al., 2011; Heck et al., 2016), and the simulation work based on HPC-EPIC was calibrated systematically but lacked independent validation and further model inter-comparisons (Kang et al., 2014).

2) To the best of our knowledge, the effect of irrigation on the yield of both Miscanthus and switchgrass, particularly their water use efficiency (WUE) in different climate zones at the global scale, has not been well studied.

3) Simulations have generally been performed for either Miscanthus (Clifton-Brown et al., 2000, 2004; Hastings et al., 2009) or switchgrass (Kang et al., 2014), with only few studies of both (Trybula et al., 2015; Ojeda et al., 2017; Beringer et al., 2011; Li et al., 2018b; Miguez et al., 2012).

4) Previous studies have generally been conducted at the regional or continental scale (Clifton-Brown et al., 2000, 2004; Hastings et al., 2009; Trybula et al., 2015; Ojeda et al., 2017); few have been conducted at the global scale (Beringer et al., 2011; Heck et al., 2016; Kang et al., 2014; Li et al., 2018b).

5) Despite their importance, the key parameters for Miscanthus and switchgrass and their differences have only been well documented in a few studies (Trybula et al., 2015).

6) Except for the LPJmL model, few models can simulate the bioenergy crop yield with full consideration of human activities in the water sector, such as irrigation, reservoir operation, and water withdrawal.

**This study well filled such gaps and limitations by enhancing and validating the H08 global hydrological model with human activities. This is realized through much efforts like collecting more observed or simulated data, transparent parameters tuning, clear parameters setting, and so on. Compared with earlier studies, our study made several important improvements as below:**

1) **Rather than using an approximation for C4 grass to represent Miscanthus and switchgrass in the LPJmL model, our enhanced H08 model simultaneously simulated the yields for Miscanthus and switchgrass at the global scale.**

2) **We conducted both site-specific calibration and independent country-specific evaluation with more observed data (as can be seen in Table 3) and predicted data (from the MISCANMOD, HPC-EPIC, and LPJmL models).**

3) **Because of the importance of transparent parameter selection as underscored by Trybula et al. (2015), we disclosed the parameters set for both Miscanthus and switchgrass.**

4) **We further investigated the differences in yield, water consumption, and especially WUE for both Miscanthus and switchgrass among different climate zones.**

5) **Except for existing models such as the LPJmL model, our enhanced H08 model provides new ways to evaluate the future impacts of human activities, such as irrigation, reservoir operation, and water withdrawal, on the large-scale establishment of bioenergy plantations.**

**In summary, we firmly believe this study is interesting (as noted by SC1), well organized (as noted by RC1) and valuable for publication in Geoscientific Model Development.**

*[R2-M1] This ms considers only two lignocellulosic herbaceous energy crops, switchgrass and Miscanthus. These species are mainly considered as energy crops for temperate and continental climate zone. For the global simulations of second-generation herbaceous energy*

*crops, authors need to address additional species in the modeling like Napier grass, sugarcane/energycane which are suitable for growing in tropics (Surendra et al. 2018). Also, consideration of woody species like Eucalyptus in tropics and SRC in boreal is recommended in the analysis of yield comparison with LPJmL and/or ORCHIDEE-MICT-BIOENERGY models as they already simulate.*

**Response: Thanks for your suggestion. We fully understand the importance of bioenergy crops you kindly mentioned. Although important, we would like to keep focusing on two herbaceous energy crops due to the following three reasons. First, Miscanthus and switchgrass have been widely targeted in many previous studies, in particular, global and continental studies (such as Clifton-Brown et al., 2000, 2004; Beringer et al., 2011; Heck et al., 2018; Kang et al., 2014; Trybula et al., 2015). Second, systematic model-intercomparison of bioenergy crops have not been provided earlier. Third, as for global models, no model can explicitly estimate the yield of Miscanthus and switchgrass. For instance, LPJmL which is one of the leading models in this field, estimates C4 grass to represent Miscanthus or switchgrass. Other models (MISCANMOD, HPC-EPIC…) can either simulate Miscanthus or switchgrass. Our enhanced H08 model simultaneously simulated the yields for Miscanthus and switchgrass at the global scale which is the good advantage to earlier studies.**

*[R2-M2] Many studies show sensitivities on the yields, and requirement, even limited amount, of N and/or P fertilizer for these crops, particularly on switchgrass (e.g. Wullschleger et al. 2010, Hong et al. 2014, Ashworth et al. 2016). SWAT model already contains implementations of the process of fertilizers applications. This is critical point.*

**Response: Thanks for your comment. We do agree with the reviewer that N and P affects the growth of the bioenergy crops. Nutrient dynamics is influenced by the complex site-specific soil conditions (soil type, temperature, wetness, carbon, etc.) which is still quite challenging to properly represent in global models. We have clearly noted this limitation in sections 3.6. We assume current yield refers to the ideal yield without constrains due to nutrients management. Similar assumption and limitation are seen in latest bioenergy potential studies**

**(Yamagata et al., 2018; Wu et al., 2019). Therefore, the results of this study can be interpreted as optimistic in the sense that N/P are sufficiently applied particularly in low-income countries and at the same time, it may not be the issue in middle or high-income countries. At least, historical validation shows without N/P consideration well reproduces (not biased).**

*[R2-M3] The simulated yields by the HO8 should be compared with simulation using SWAT model with the same climate forcing to validate the adopted implementations from the model described in the ms.*

**Response: Thanks for your comment. Although interesting, we haven't proceeded this since it is always hard to interpret the results of model-model comparison. Alternatively, we have provided the solid calibration/validation results (i.e. model-observation comparison) in Fig. 3 and Fig. 4. Moreover, we have made comparison with other three models as shown in Fig. 5. The result is comparable with other global models like HPC-EPIC and LPJml. Please note that a global application of SWAT model is quite laborious hence there are no available global bioenergy yield data from SWAT at present.**

*[R2-M4] The simulation uses climatology (1997-2016 average) for the input, however, to validate the model results with field observations, I would recommend using annual forcing data in the simulations for the corresponding years. In addition, only WFDEI forcing is used in the simulations. There is no supporting information on the rational of using the forcing date over other data set, like S14FD (Iizumi et al. 2017), CRU JRA. (Harris et al. 2019). At minimum, a revised manuscript must address the limitations.*

**Response: Thanks for your comment. First, we took your suggestion and conducted simulations with annual forcing meteorological data and revised the text and figures in the revised paper. The main results and conclusion did not change greatly. Regarding the selection of meteorological data, we also added the simulation results with meteorological driven from S14FD (Iizumi et al. 2017), as can be find in Fig. s2, the results are very similar with current driven form WFDEI. Therefore, we kept our results with driven from WFDEI data.**

*[R2-M5] It is critically needed to compared/validate the simulated yields using irrigation with field observations/experiments if data would be available.*

**Response: Thanks for your comment. We have added the result under irrigated condition with 10 available site observations as shown in Fig. s3. As can be seen, both good and bad result exist. However, it is difficult to judge the performance due to the limited number of observations. Moreover, we also added the comparison results with LPJml model under irrigation condition. The result is plotted in Fig. s4. It shows our simulation is consistent with LPJml. The correlation coefficient of the yield simulated by H08 and LPJmL in the scatter plot (Fig. s4) was 0.95. A t-test showed that the correlation was significant at the 0.01 level.**

*Minor comments:*

*[R2-S1] L16: Bauer et al. 2018 could be a "recent predictions".*

**Response: Thanks for recommending this paper. We added it in the revised manuscript.**

*[R2-S2] L61: Miguez et al. 2012 also considers both species.*

**Response: Thanks for recommending this paper. We added it in the revised manuscript.**

*[R2-S3] L62-64: Is it really few studies available? Nair et al. 2012 reviews and shows many models already available for the simulation*

**Response: First, thanks for recommending this review paper. In total, 14 models were introduced in the paper. As clearly documented in the paper, most of them are for field scale. Only two models called Agro-IBIS and LPJmL are for regional/global scale. Therefore, it means few simulations have been done at global stale, as we originally stated in Line 62-64.**

*[R2-S4] L154: I found no description on simulation algorithm of parameter ensemble in the supporting materials.*

**Response: We offered specific equations containing the parameters in the supplementary file. For example, base temperature (Tb) can be found in equation 1, the potential heat unit (Hun) can be found in equation 2, and so forth. Regarding the parameter ensemble in the simulation is summarized in Table 2. We conducted simulations with all combinations of each parameter as documented in Table 2.**

*[R2-S5] Fig.3: Is it possible to disaggregate the figure with each climate zones?*

**Response: Thanks for your comment, and we have identified the climate zone for each site and modified Fig. 3.**

*[R2-S6] Table s1, s2: I would recommend adding mean and SD of the observed yields in the tables.*

**Response: Thanks for your suggestion. We added the mean yield and the range of minimum and maximum yield in Table s1 and s2.**

*[R2-S7] L190: Too small sample size (N = 8).*

**Response: There are 15 countries' yield reported in MISCANMOD, originally, we made comparison for the 8 counties where yield simulated by H08 meet the requirement that higher than 10 Mg ha$^{-1}$yr$^{-1}$. In the revision, we used all the 15 samples in the comparison with the modification mentioned in response to [R2-S8] below.**

*[R2-S8] L192: I don't understand the reason of excluding that yield below 10 Mg ha$^{-1}$ yr$^{-1}$.*

**Response: The reason is to make the comparison meaningful since MISCANMOD reported the results exceeding 10 Mg ha$^{-1}$yr$^{-1}$ (Clifton-Brown et al., 2004). In the revision, we take more accurate method that consistent with MISCANMOD reports (Clifton-Brown et al., 2004) by excluding the area within a country if the yield is below 10 Mg ha$^{-1}$yr$^{-1}$ before the country-average calculation. In addition, we also changed the calculation period from 1979-2016 to 1979-1990 to make it more consistent with MISCANMOD reports.**

*[R2-S9] L199: Does the result of Fig 5g change if the higher yield from Miscanthus or switchgrass calculated by H08 are used in the comparison?*

**Response: Thanks for your comment, we have made new figure (Fig. s5) with the separated yield of Miscanthus or switchgrass from H08 in supplementary file.**

*[R2-S10] L203: Is the discrepancy related to the modeling assumption of double cropping in the region? In practice, double or triple cropping can be done in the tropics for the herbaceous crops.*

**Response: Thanks for your suggestion. There are no assumptions of double cropping in those tropical zones in LPJml and H08.**

*[R2-S11] L215: I recommend comparing simulated results between SWAT and H08 using same forcing data.*

**Response: Thanks for your comment. As we responded in the [R2-M3], although interesting, let us drop this since model-model inter-comparison is hard to interpret.**

**References**

**Beringer, T. I. M., Lucht, W., and Schaphoff, S.: Bioenergy production potential of global biomass plantations under environmental and agricultural constraints, GCB Bioenergy, 3, 299-312, https://doi:10.1111/j.1757-1707.2010.01088.x, 2001.**

**Bondeau, A., Smith, P. C., Zaehle, S., Schaphoff, S., Lucht, W., Cramer, W., Gerten, D., Lotze-Campen, H., Müller, C., Reichstein, M., and Smith, B.: Modelling the role of agriculture for the 20th century global terrestrial carbon balance, Glob. Change Biol., 13, 679-706, https://doi:10.1111/j.1365-2486.2006.01305.x, 2007.**

**Clifton-Brown, J. C., Neilson, B., Lewandowski, I., and Jones, M. B.: The modelled productivity of Miscanthus×giganteus (GREEF et DEU) in Ireland, Ind. Crops Prod., 12, 97-109, https://doi:10.1016/S0926-6690(00)00042-X, 2000.**

Clifton-Brown, J. C., Stampfl, P. F., and Jones, M. B.: Miscanthus biomass production for energy in Europe and its potential contribution to decreasing fossil fuel carbon emissions, Glob. Change Biol., 10, 509-518, https://doi:10.1111/j.1529-8817.2003.00749.x, 2004.

Heck, V., Gerten, D., Lucht, W., and Boysen, L. R.: Is extensive terrestrial carbon dioxide removal a 'green' form of geoengineering? A global modelling study, Glob. Planet. Change, 137, 123-130, https://doi:10.1016/j.gloplacha.2015.12.008, 2016.

Kang, S., Nair, S. S., Kline, K. L., Nichols, J. A., Wang, D., Post, W. M., Brandt, C. C., Wullschleger, S. D., Singh, N., Wei, Y.: Global simulation of bioenergy crop productivity: analytical framework and case study for switchgrass, GCB Bioenergy, 6, 14-25, https://doi:10.1111/gcbb.12047, 2014.

Trybula, E. M., Cibin, R., Burks, J. L., Chaubey, I., Brouder, S. M., and Volenec, J. J.: Perennial rhizomatous grasses as bioenergy feedstock in SWAT: parameter development and model improvement, GCB Bioenergy, 7, 1185-1202, https://doi:10.1111/gcbb.12210, 2015.

Yamagata, Y., Hanasaki, N., Ito, A., Kinoshita, T., Murakami, D., and Zhou, Q.: Estimating water–food–ecosystem trade-offs for the global negative emission scenario (IPCC-RCP2.6), Sustainability Sci., 13(2), 301-313, https://doi:10.1007/s11625-017-0522-5, 2018.

Hastings, A., Clifton-Brown, J., Wattenbach, M., Mitchell, C. P., and Smith, P.: The development of MISCANFOR, a newMiscanthuscrop growth model: towards more robust yield predictions under different climatic and soil

conditions, GCB Bioenergy, 1, 154-170, https://doi:10.1111/j.1757-1707.2009.01007.x, 2009.

Li, W., Yue, C., Ciais, P., Chang, J., Goll, D., Zhu, D., Peng, S., and Jornet-Puig, A.: ORCHIDEE-MICT-BIOENERGY: an attempt to represent the production of lignocellulosic crops for bioenergy in a global vegetation model, Geosci. Model Dev., 11, 2249-2272, https://doi.org/10.5194/gmd-11-2249-2018, 2018b.

Ojeda, J. J., Volenec, J. J., Brouder, S. M., Caviglia, O. P., and Agnusdei, M. G.: Evaluation of Agricultural Production Systems Simulator as yield predictor ofPanicum virgatumandMiscanthusxgiganteusin several US environments, GCB Bioenergy, 9, 796-816, https://doi:10.1111/gcbb.12384, 2017.

Miguez, F. E., Maughan, M., Bollero, G. A., and Long, S. P.: Modeling spatial and dynamic variation in growth, yield, and yield stability of the bioenergy crops Miscanthus×giganteus and Panicum virgatum across the conterminous United States. GCB Bioenergy, 4(5), 509-520, https://doi.org/10.1111/j.1757-1707.2011.01150.x, 2012.

Iizumi, T., Takikawa, H., Hirabayashi, Y., Hanasaki, N., and Nishimori, M.: Contributions of different bias-correction methods and reference meteorological forcing data sets to uncertainty in projected temperature and precipitation extremes. Journal of Geophysical Research: Atmospheres, 122(15), https://doi.org/10.1002/2017JD026613, 7800-7819, 2017.

Thank you very much for your comments and suggestions.
Sincerely yours,
Zhipin Ai (on behalf of the co-authors)

---

## Author Comment (AC3) · 29 Jan 2020

We would like to thank Dr. Liyin He for the positive comments to the manuscript. The reviewer's comment was indicated in italic and the response was indicated in bold letters.

*#SC1*

*[S1-M1] I am working on using satellite remote sensing to estimate the crop yield. I am interested in this paper since it aims to improve the hydrologic model to estimate the bioenergy crop yields on the global scale. Although I am new to crop models, I feel this paper is easy to understand and provide some insights for my own work. I would appreciate authors share more details about the model parameters and compare simulation results with past models. Then, the audiences will have more direct feelings about the improvement of the model*

**Response: Thanks for your interest and comments on our paper. The details on parameters are presented in Table 1 and 2, and the text in section 3.1. Detailed explanations and equations related to the parameters can be found in the supplementary file. We also added more comparison results under irrigated condition in the supplementary file. In addition, you can learn more information about the parameters in H08 from the paper Hanasaki et al. (2008) and the manual on the website (http://h08.nies.go.jp/h08/manual.html).**

**Hanasaki, N., Kanae, S., Oki, T., Masuda, K., Motoya, K., Shirakawa, N., et al. (2008a). An integrated model for the assessment of global water resources—Part 2: Applications and assessments. Hydrology and Earth System Sciences, 12(4), 1027–1037. https://doi.org/10.5194/hess-12-1027-2008**

**Thank you very much for your comments.**

**Sincerely yours,**

**Zhipin Ai (on behalf of the co-authors)**